# ACTION CHUNKING PROXIMAL POLICY OPTIMIZATION FOR UNIVERSAL DEXTEROUS GRASPING

## ABSTRACT

Universal dexterous grasping across diverse objects is a crucial step towards human-like manipulation. In order to handle the high degrees of freedom (DoF) of dexterous hands, state-of-the-art universal dexterous grasping methods adopt on-line reinforcement learning (RL) algorithms such as Proximal Policy Optimization (PPO) to learn action policies. Although PPO is a common choice, its vanilla version often leads to insufficient exploration and slow policy improvement, requiring additional training augmentation to achieve high performance. While action chunking is a promising strategy to improve exploration by temporally coherent actions, prior RL algorithms that integrate action chunking are unsuitable for dexterous hands due to their high-DoF Q-functions. To address this, we reformulate the PPO objective over action chunks and use a standard state-value function as the critic, naming *Action Chunking Proximal Policy Optimization* (ACPPO). ACPPO retains the simplicity of PPO while encouraging temporally coherent exploration and avoiding the curse of dimensionality. Validating on the DexGraspNet dataset, we observe that ACPPO outperforms all prior PPO-based methods by a success rate of 95.4% with $2.3\times$ faster training without any auxiliary learning mechanisms.

## 1 INTRODUCTION

Dexterous hands (Kappassov et al., 2015; Iberall, 1997; Pons et al., 1999; Sampath et al., 2023) are robotic grippers that emulate the versatility of the human hand. While achieving human-level dexterity would unlock a whole new field of applications, dexterous grasping (Kumar et al., 2016; Ciocarlie et al., 2007; Li et al., 2016) still remains a challenge due to the high degrees of freedom (DoF) and diversity of the target objects.

Recent works (Wan et al., 2023; Zhang et al., 2024a; Huang et al., 2025; Wang et al., 2025) utilize reinforcement learning (RL) (Sutton et al., 1998), especially Proximal Policy Optimization (PPO) (Schulman et al., 2017) to learn universal dexterous grasping across diverse objects, varying in size and geometry (Xu et al., 2023). However, while PPO itself has proven to be excellent in robotic environments with small DoF (Raffin, 2020; Raffin et al., 2021; Saeed et al., 2021; Han et al., 2023), the algorithm alone struggles in the high-DoF dexterous grasping environment. Therefore, recent works augment PPO with additional mechanisms such as curriculum learning (Wan et al., 2023), residual learning and mixture of experts (Huang et al., 2025), transformers (Wang et al., 2025), or motion objectives (Zhang et al., 2024a).

One promising direction for universal dexterous grasping is action chunking reinforcement learning (ACRL), where the policy outputs a short sequence of actions at each decision point. By committing multi-step commands, action chunking encourages temporally coherent exploration, exploring a more diverse set of states. (Li et al., 2025b). This provides the critic with better knowledge of the environment, which improves the performance of the learned policy.

Despite its potential, no prior ACRL method has been successfully applied to high-DoF dexterous grasping since existing designs are fundamentally incompatible with the scale of the problem. Prior action chunking methods learn a chunked action-value function $Q(s_t, a_{t:t+h-1})$ (Li et al., 2025a;b; Seo & Abbeel, 2024), where the action space dimension $h|\mathcal{A}|$ becomes impractical for dexterous hands with 20+ DoF (Ding et al., 2024).

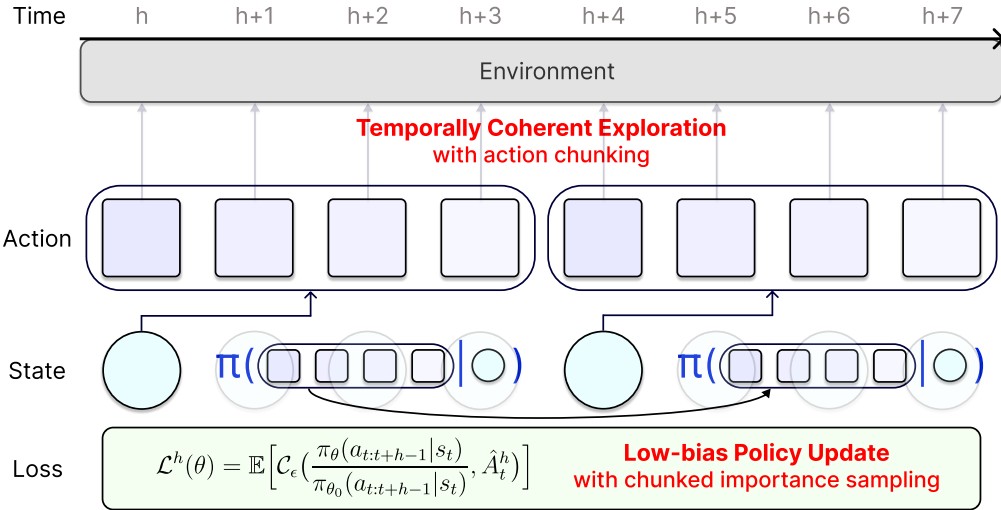

Figure 1: The overview of our method, Action Chunking Proximal Policy Optimization (ACPPO). Using a chunked actor, ACPPO improves exploration with temporally coherent actions. Furthermore, the loss term with chunked importance sampling reduces bias when incorporating Generalized Advantage Estimates, making each policy update more stable and efficient.

Here, we propose **Action Chunking Proximal Policy Optimization (ACPPO)**, the first method to apply action chunking in the domain of dexterous grasping. As highlighted in Fig. 1, our key insight is to integrate the action chunking mechanism directly into the actor and the on-policy surrogate objective, while completely avoiding intractable chunked Q-functions. Instead, ACPPO uses a simple, chunk-size independent state-value critic $V(s)$, allowing temporally coherent exploration and a lower surrogate bias while preserving the simplicity of PPO.

On the DexGraspNet dataset (Wang et al., 2023), ACPPO establishes a new state-of-the-art success rate of $95.4\%$, outperforming prior PPO-based methods. Furthermore, ACPPO trains $2.3\times$ faster, as it does not require additional training mechanisms.

Our main contribution can be summarized as follows:

- We propose Action Chunking PPO, the first state-value function based action chunking reinforcement learning algorithm designed for high-dimensional continuous control tasks like dexterous grasping.

- We reformulate PPO over action chunks and derive a clipped surrogate with chunked importance ratio, encouraging temporally coherent exploration and reducing the bias from Generalized Advantage Estimates.

- We demonstrate that ACPPO outperforms all PPO-based methods both in success rate and training efficiency in the DexGraspNet dataset, establishing a strong baseline for universal dexterous grasping.

## 2 RELATED WORK

### 2.1 DEXTEROUS GRASPING

Dexterous Grasping (Okamura et al., 2000; Mukherjee, 1992) is a formidable challenge in robotics, due to its high DoF and complex geometries of the target objects. While the initial challenge is to generate an practical grasping pose for each object, recent works propose various methods such as affordance (Yang et al., 2024; Mandikal & Grauman, 2021; Zurbrügg et al., 2025), grasping datasets (Zhong et al., 2025; Chen et al., 2025; Wang et al., 2023), or diffusion models (Weng et al., 2024; Zhang et al., 2024b).

Recently, reinforcement learning has been adopted in the domain of dexterous grasping since it can learn high Degree-of-Freedom (DoF) action policies without requiring explicit access to the environmental dynamics. One major direction of exploiting reinforcement learning is focused on imitation learning (Qin et al., 2022; Arunachalam et al., 2022; An et al., 2025), where policies are trained to mimic expert demonstrations. These methods focus on reducing distribution shift, and collecting demonstrations efficiently, for example through human videos.

Another line of work focuses on on-policy learning, where the policy is updated with its own rollouts. Our universal dexterous manipulation setup is mainly part of this group, since it is costly to collect demonstrations for thousands of different objects. Proximal Policy Optimization (PPO) (Schulman et al., 2017) is widely used in this setting, since it offers stable training with clipped surrogate objectives. While PPO-based methods have shown promising results in universal dexterous grasping, they typically rely on additional mechanisms such as curriculum learning (Wan et al., 2023), residual learning (Huang et al., 2025), or transformer backbones (Wang et al., 2025). This is primarily due to the high-dimensional action space, which leads to challenging exploration and optimization of the policy.

## 2.2 ACTION CHUNKING

Action chunking is a method that predicts and executes a sequence of actions rather than single-step commands (Zhao et al., 2023). In imitation learning, action chunking has been known to improve the robustness of learned policies and handle non-Markovian behaviors of human demonstrations (George & Farimani, 2023; Bharadhwaj et al., 2024). Recent work integrated action chunking into reinforcement learning, reporting that action chunking encourages temporally coherent exploration and leads to efficient critic learning. (Li et al., 2025a;b; Seo & Abbeel, 2024).

However, these designs are impractical for dexterous grasping. Prior action chunking RL methods focus on learning the chunked action value function $Q(s_t, a_{t:t+h-1})$ with chunk length $h$. The action input dimension $\mathbb{R}^{h|\mathcal{A}|}$ is tractable for low-DoF robots such as manipulators (DoF $\leq 7$) but becomes unmanageable for dexterous hands with 20+ DoF (Ding et al., 2024). In addition, several action chunking RL methods (Li et al., 2025b; Seo & Abbeel, 2024) rely on an offline dataset, which is intractable for universal grasping. Universal grasping inevitably requires high-dimensional visual features of the target objects, making it extremely costly to cover the state space as offline datasets. Architecturally, the transformer backbones used in some prior work (Li et al., 2025a) add expressibility to the chunked action value function, but also become computationally costly as the action dimension increases in dexterous hand settings.

To address these limitations, we propose Action Chunking Proximal Policy Optimization (ACPPO), a fully online action-chunked RL method that operates by learning the state-value function $V(s)$. By avoiding the high-dimensional chunked $Q$ function, we mitigate the effect of $\mathbb{R}^{h|\mathcal{A}|}$ while retaining the benefits of action chunking.

## 3 PRELIMINARIES

### 3.1 PROBLEM STATEMENT

To address the universal dexterous grasping task, we consider a Markov Decision Process (MDP) (Sutton et al., 1998). It is defined by the tuple $(\mathcal{S}, \mathcal{A}, p, r, \gamma)$, where $\mathcal{S}$ is the state space, $\mathcal{A}$ is the action space, $p(s_{t+1}|s_t, a_t) : \mathcal{S} \times \mathcal{A} \to \mathcal{S}$ is the transition probability function, $r(s, a) : \mathcal{S} \times \mathcal{A} \to \mathbb{R}$ is the reward function, and $\gamma \in [0, 1)$ is the discount factor. The behavior of the agent is determined by a stochastic policy $\pi(a|s)$.

For the hand model, we use Shadow Hand (Shadow Robot, 2025). The states are concatenations of robot proprioception $\in \mathbb{R}^{167}$, actions of the last step $\in \mathbb{R}^{24}$, pose of the object and the goal $\in \mathbb{R}^{16}$, and PointNet features (Qi et al., 2017) of the target object $\in \mathbb{R}^{64}$. The exact input state is described in Table 1. The action specifies the 24 DoF input to the Shadow Hand, containing 6 for the wrist, 5 for the thumb, 4 for the little finger, and 3 for each of the remaining fingers. Following Huang et al. (2025), the reward function is designed as:

$$R = R_{\text{dist}} + R_{\text{align}} + f_{\text{contact}} \left( R_{\text{goal}} + R_{\text{lift}} + R_{\text{bonus}} \right). \tag{1}$$

| Input Type | Elements (Dimension) |
|---|---|
| Proprioception (167) | Wrist pose (6); Finger joints (angle, angular velocity, force) (22×3); Fingertip position (5×3), quaternion rotation (5×4), linear velocity (5×3), angular velocity (5×3), force (5×3) and torque (5×3). |
| Previous Action (24) | Wrist force (3) and torque (3); Finger-joint angles (18). |
| Object State (16) | Object pose (7), linear velocity (3), and angular velocity (3); Object-goal distance (3). |
| Object Feature (64) | PointNet features (64). |

Table 1: Input state for the policy network. We consider a total 273D state input as a concatenation of robot proprioception, previous action, object and goal states, and object features.

Here, $R_{\text{dist}}$ penalizes the object-hand distance, while $R_{\text{align}}$ encourages a wide finger spread for the stability of grasping. The remaining rewards are gated by a binary flag $f_{\text{contact}}$, which activates in contact between the hand and the object. In contact, $R_{\text{goal}}$ penalizes the object-goal distance, $R_{\text{lift}}$ encourages the hand to lift up the object, and $R_{\text{bonus}}$ is a bonus term when the object is placed near the goal. Further descriptions on the reward function can be found in Appendix B.1.

## 3.2 PROXIMAL POLICY OPTIMIZATION

Given a batch of trajectories collected by the behavior policy $\pi_{\theta_0}$ (typically the policy from the last update), PPO (Schulman et al., 2017) performs multiple epochs of first-order updates on a clipped surrogate objective that controls the deviation of $\pi_\theta$ from $\pi_{\theta_0}$.

Define the clipping minimum operator $\mathcal{C}_\epsilon$ and the per-step importance sampling ratio $\rho_t(\theta)$ as:

$$\mathcal{C}_\epsilon(\rho, A) := \min\left(\rho A, \text{clip}_{1-\epsilon}^{1+\epsilon}(\rho)A\right), \qquad \rho_t(\theta) := \frac{\pi_\theta(a_t|s_t)}{\pi_{\theta_0}(a_t|s_t)}, \tag{2}$$

where $\text{clip}_{1-\epsilon}^{1+\epsilon}(\rho) := \max(1-\epsilon, \min(\rho, 1+\epsilon))$.

Let $\hat{A}_t$ denote an estimate of the advantage $A^{\pi_{\theta_0}}(s_t, a_t)$ via Generalized Advantage Estimates (GAE) (Schulman et al., 2015b). PPO maximizes the clipped surrogate

$$\mathcal{L}_{\text{PPO}}(\theta) := \mathbb{E}_{\pi_{\theta_0}}\left[\mathcal{C}_\epsilon\left(\rho_t(\theta), \hat{A}_t\right)\right]. \tag{3}$$

This keeps $\rho_t$ inside the trust region $[1-\epsilon, 1+\epsilon]$, following the behavior of TRPO (Schulman et al., 2015a) while keeping a simple first-order update. In practice, PPO is implemented with multiple optimization epochs per batch and often includes entropy regularization. The full details of PPO are provided in A.1.

## 4 METHOD

While effective across a wide range of domains (Raffin, 2020; Xiao, 2023; Zheng et al., 2023), PPO operates on step-wise action advantages, causing insufficient exploration in high-DoF tasks such as dexterous manipulation. This motivates our method, Action Chunking PPO. Our primary challenge is to reformulate the objective over action chunks while keeping the relative objective function unbiased, and maintaining a stable importance sampling ratio.

## 4.1 ACTION CHUNKING PPO

For merging action chunking into the domain of reinforcement learning, we define a chunked actor $\pi(a_{t:t+h-1}|s_t)$ that predicts the next $h$ actions from the current state. Also, we define the chunked advantage as follows, which is intuitively the sum of advantages inside the chunk:

$$A^{\pi_{\theta_0}}(s_t, s_{t+h-1}) := \sum_{k=0}^{h-1} \gamma^k r_{t+k} + \gamma^h V^{\pi_{\theta_0}}(s_{t+h}) - V^{\pi_{\theta_0}}(s_t). \tag{4}$$

---

**Algorithm 1** Action-Chunked PPO (ACPPO; boundary-GAE variant)

---

**Inputs:** horizon $h$, clip $\epsilon$, LR $\alpha$, policy $\pi_\theta$, value $V_\phi$

  **while** not converged **do**

    **for rollout** steps $t = 0, \ldots, T - 1$ **do**

      **if** $t \bmod h = 0$ **then**

        Sample action-chunk $\mathbf{a}_{t:t+h-1} \sim \pi_\theta(\cdot \mid s_t)$

        Cache $\log \pi_{\theta_0}(\mathbf{a}_{t:t+h-1} \mid s_t)$ and boundary states $(\mu, \log \sigma)$

      Execute $a_t$, store $(s_t, a_t, r_t, d_t, V_\phi(s_t))$

    Compute returns $\hat{R}_t$ and step-wise advantages $\hat{A}_t^{\mathrm{GAE}(\lambda)}$

    **for epochs** $e = 1, \ldots, E$ **do**

      Sample mini-batches of timesteps $\mathcal{B}$

      Restrict policy terms to chunk boundaries $\mathcal{B}_{\mathrm{policy}} = \mathcal{B} \cap \partial\mathrm{chunk}$

      Compute joint ratio $\rho_{t,h}^{\mathrm{ch}}(\theta) = \exp\big(\log \pi_\theta(\mathbf{a}_{t:t+h-1} \mid s_t) - \log \pi_{\theta_0}(\mathbf{a}_{t:t+h-1} \mid s_t)\big)$

      **Policy:** Maximize $\mathbb{E}_{t \in \mathcal{B}_{\mathrm{policy}}}\left[\mathcal{C}_\epsilon\big(\rho_{t,h}^{ch}(\theta), \hat{A}_t^{\mathrm{GAE}(\lambda)}\big)\right]$

      **Value:** Minimize $\mathbb{E}_{t \in \mathcal{B}}\left[\big(V_\phi(s_t) - \hat{R}_t\big)^2\right]$

      Adapt $\alpha$ using chunk KL $D_{\mathrm{KL}}\big[\pi_{\theta_0}(\cdot \mid s_t) \,\|\, \pi_\theta(\cdot \mid s_t)\big]$

      Update $(\theta, \phi)$ with LR $\alpha$

    Update behavior policy: $\pi_{\theta_0} \leftarrow \pi_\theta$

---

Since the actor outputs a length $h$ sequence of actions, the importance sampling ratio now becomes:

$$\rho_{t,h}^{ch}(\theta) := \frac{\pi_\theta(a_{t:t+h-1} | s_t)}{\pi_{\theta_0}(a_{t:t+h-1} | s_t)}, \tag{5}$$

Using these formulas, the relative objective function for the policy gradient can be written as:

$$\mathcal{J}(\theta) - \mathcal{J}(\theta_0) = \mathbb{E}_{\substack{\tau \sim (p_0, \pi_\theta, p) \\ a_t' \sim \pi_{\theta_0}}} \left[ \sum_{t=0}^{T-1} \gamma^t \frac{\pi_\theta(a_t' | s_t)}{\pi_{\theta_0}(a_t' | s_t)} A^{\pi_{\theta_0}}(s_t, a_t') \right] \tag{6}$$

$$= \mathbb{E}_{\substack{\tau \sim (p_0, \pi_\theta, p) \\ a_t' \sim \pi_{\theta_0}}} \left[ \sum_{l=0}^{(T-1)/h} \gamma^{lh} \frac{\pi_\theta(a_{t:t+h-1} | s_t)}{\pi_{\theta_0}(a_{t:t+h-1} | s_t)} A_h^{\pi_{\theta_0}}(s_{lh}, s_{l(h+1)-1}) \right]. \tag{7}$$

Eq. 6 is reduced into the relative objective function of standard PPO when $h = 1$. Following the standard PPO methodology, we construct a surrogate objective function by sampling on $\pi_{\theta_0}$, then apply clipping for stable policy updates:

$$\mathcal{L}_{\mathrm{ACPPO}}^h(\theta) := \mathbb{E}_l\left[\mathcal{C}_\epsilon\big(\rho_{lh,h}^{ch}(\theta), A_h^{\pi_{\theta_0}}(s_{lh}, s_{l(h+1)-1})\big)\right]. \tag{8}$$

For further reducing variance, we replace the chunked advantage with a Generalized Advantage Estimate (GAE), defined with temporal-difference residuals $\delta_t$:

$$\delta_t^{(\phi)} := r_t + \gamma V_\phi(s_{t+1}) - V_\phi(s_t), \tag{9}$$

$$\hat{A}_t^{\mathrm{GAE}(\lambda)} := \sum_{l=0}^{\infty} (\gamma\lambda)^l \, \delta_{t+l}. \tag{10}$$

This yields our final objective function of ACPPO:

$$\mathcal{L}_{\mathrm{ACPPO}}^h(\theta) := \mathbb{E}_t\left[\mathcal{C}_\epsilon\big(\rho_{t,h}^{chunk}(\theta), \hat{A}_t^{\mathrm{GAE}(\lambda)}\big)\right]. \tag{11}$$

While the application of GAE introduces additional bias into the surrogate loss term, this bias is smaller compared to the case of PPO. With a chunk size of $h$, the GAE can be decomposed as:

$$\hat{A}_t^{\mathrm{GAE}(\lambda)} = \underbrace{\sum_{j=0}^{h-1} (\gamma\lambda)^j \, \delta_{t+j}}_{\text{inside chunk}} + \underbrace{\sum_{j=h}^{\infty} (\gamma\lambda)^j \, \delta_{t+j}}_{\text{tail bias}}. \tag{12}$$

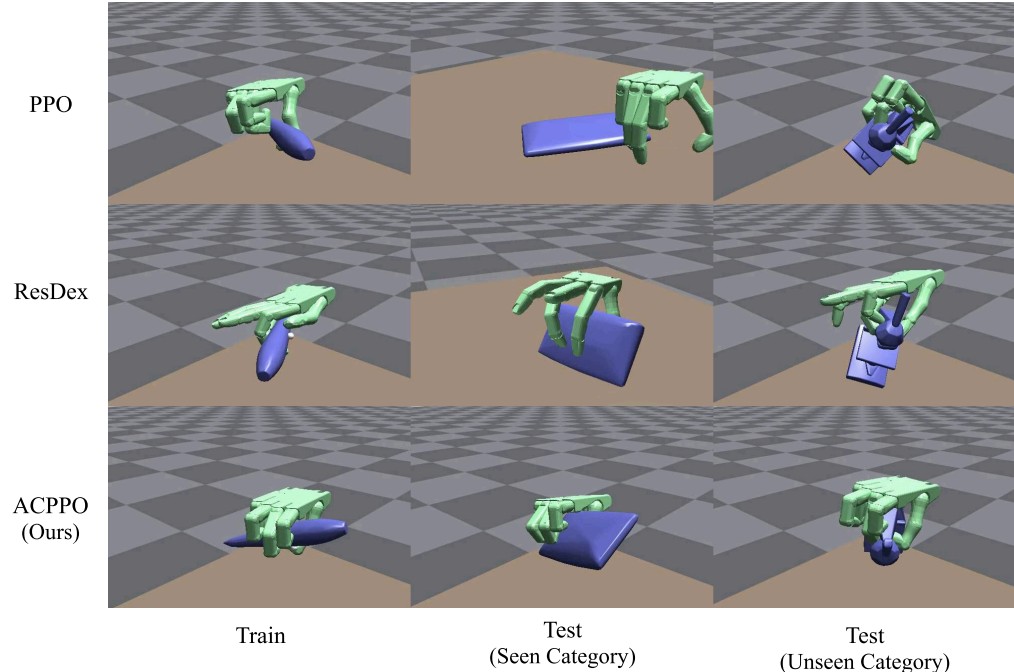

Train     Test (Seen Category)     Test (Unseen Category)

Figure 2: Qualitative grasping pose comparisons between PPO (Schulman et al., 2017) (top), Res-Dex (Huang et al., 2025) (middle), and ACPPO (bottom) across train, test-seen category, test-unseen category objects. ACPPO consistently outputs more stable grasp poses compared to prior methods.

In PPO, the tail bias term starts at $j = 1$, while for ACPPO, the tail bias term starts at $j = h(> 1)$. This suppresses the bias for ACPPO by a factor of $(\gamma\lambda)^{h-1}$ relative to PPO, yielding a more accurate advantage estimate.

The ACPPO training loop is summarized in Algorithm 1, and the complete derivation of ACPPO is clarified in A.2.

## 4.2 BENEFITS OF ACTION CHUNKING IN DEXTEROUS MANIPULATION

By training a policy to output action sequences, ACPPO provides several key advantages that are particularly impactful for high-DoF dexterous manipulation tasks. The core benefit, which has also been observed by previous action chunking studies (Li et al., 2025b; Zhao et al., 2023; George & Farimani, 2023; Bharadhwaj et al., 2024), is temporally coherent exploration.

Standard on-policy training often begins with high-frequency, jittery motions. Because these small random actions cancel each other out, this form of exploration effectively traps the agent within a small, local domain of the state space. Action chunking encourages the execution of smoother, multi-step movements, allowing the policy to explore a much larger portion of the vast state space of the high DoF dexterous hand. While this broader exploration may result in a slower discovery of the global optimum direction, it allows the policy to gain a more comprehensive understanding of the environment's dynamics. This foundation leads the policy to build a better grasp pose in the long term, yielding higher final performance.

Beyond exploration, the chunked actor also benefits in inference. First, it improves inference speed by reducing the frequency of policy forward passes by $h$. Furthermore, the action chunks act like a low-pass filter on the commanded motion: when only part of the dexterous hand (e.g. one or two fingers) makes early contact with the objects, single-step policies often overreact with large whole-hand corrections. However, the chunked policy, on the other hand, continues the planned action sequence, avoiding oscillatory movement during rollout.

Table 2: Success rates of state-based policies on the DexGraspNet dataset. Without any additional training mechanisms, ACPPO achieves state-of-the-art performance. ACPPO also excels in training time, training 2.3× faster compared to the prior fastest method, ResDex (Huang et al., 2025).

| Method | RL method | Train (%) | Test (%) | | Training Time |
| | | | Seen | Unseen | |
| --- | --- | --- | --- | --- | --- |
| Xu et al. (2023) | PPO + Curriculum | 79.4 | 74.3 | 70.8 | -[1] |
| Wan et al. (2023) | PPO + Curriculum | 87.9 | 84.3 | 83.1 | 48h[2] |
| Wang et al. (2025) | PPO + Transformer | 91.2 | 89.2 | 88.3 | 70h[3] |
| Huang et al. (2025) | PPO + Residual | 94.6 | 94.4 | 95.4 | 16h[4] |
| ACPPO (Ours) | ACPPO | **95.4±0.1** | **94.8±0.3** | **95.6±0.4** | **7h**[4] |

[1]Training details not released.
[2]Reported value, trained on 4× NVIDIA RTX 3090 Ti GPUs.
[3]Reported value, trained on 8× NVIDIA A100 GPUs.
[4]Trained on a single NVIDIA RTX A6000 GPU.

## 5 EXPERIMENTS

### 5.1 SET-UPS

**Dataset.** We evaluate our method on the DexGraspNet dataset (Wang et al., 2023), which consists of 5519 object instances from 133 object categories. Following UniDexGrasp++ (Wan et al., 2023), we use 3200 object instances for the training set, and construct two distinct test sets: (i) Seen-category objects, containing 141 unseen instances from categories included in the training set; (ii) Unseen-category objects, containing 100 objects from novel categories.

**Implementation Details.** For training the policy with ACPPO, we construct 6,400 parallel simulation environments on IsaacGym (Makoviychuk et al., 2021). We train for 30,000 iterations on three different seeds, using a single NVIDIA RTX A6000 GPU. For testing, we run 100 rollouts for each object. A rollout is considered successful if the object is placed at the desired goal position $z > 0$ after all action sequences have been executed. For fairness, we keep the total environment steps and network sizes of the RL algorithm matched with the prior methods, Xu et al. (2023), Wan et al. (2023) and Huang et al. (2025). The hyperparameters for our training can be found in Appendix B.2.

### 5.2 RESULTS

We compare our method with the baseline methods using the policy trained with chunk size $h = 2$. The ablation study on the size of action chunks is presented in Section 6.3. The baselines include PPO combined with curriculum learning (Xu et al., 2023; Wan et al., 2023), residual learning and mixture of experts (Huang et al., 2025), and transformer backbones (Wang et al., 2025), all designed specifically for universal dexterous grasping in the DexGraspNet dataset.

Table 2 demonstrates that our method outperforms all PPO-based algorithms in universal dexterous grasping. Although the success rate improvement compared to the previous state-of-the-art model Huang et al. (2025) can be viewed as marginal, it is notable that we achieve this result without any auxiliary training methods. This simple design also leads to superior training efficiency. The fastest prior method Huang et al. (2025) requires 40,000 iterations with 11,000 parallel environments to train their hyper-policy, taking 16 hours on a single NVIDIA RTX A6000 GPU (reported as 11 hours on a single NVIDIA RTX 4090 GPU). In contrast, ACPPO reaches higher performance within only 30,000 iterations on 6,400 environments, completing training in 7 hours on the same device.

## 6 ABLATION STUDY

For ablation study, we construct 3,200 parallel environments and train for 30,000 iterations, due to the additional memory usage of double Q-policy networks (Van Hasselt et al., 2016) in Li et al. (2025b).

Table 3: Experiment results of ACFQL, PPO, and ACPPO on the DexGraspNet dataset. The policies are trained on 3200 parallel environments for 30000 iterations, PPO and ACPPO trained on 3 different seeds.

| Method | Train (%) | Test (%) | | Training Time (s) |
| --- | --- | --- | --- | --- |
| | | Seen | Unseen | |
| ACFQL (Li et al., 2025b) | 77.1 | 76.7 | 79.0 | **15,125** |
| PPO (Schulman et al., 2017) | 90.5±1.1 | 90.1±1.0 | 92.3±0.5 | 17,803 |
| ACPPO (Ours) | **93.4±1.3** | **93.1±1.4** | **94.3±0.5** | 15,441 |

## 6.1 COMPARISON TO PRIOR REINFORCEMENT LEARNING ALGORITHMS

We compare the performance of ACPPO against two RL algorithms: Action Chunking Flow Q-Learning (ACFQL) (Li et al., 2025b) and PPO (Schulman et al., 2017). PPO is the state-of-the-art on-policy method, and ACFQL is a prior action chunking RL algorithm that relies on Q-functions. For ACFQL, we used a chunk size of $h = 5$, which showed the best performance. As shown in Table 3, ACPPO consistently outperforms both ACFQL and PPO across all metrics: training success rate, and test success rates on both seen and unseen objects.

Comparing the learning dynamics of PPO and ACPPO, ACPPO improves more slowly in the early stage, but it soon overtakes PPO and converges to a higher success rate. The initial lag is due to the temporally coherent exploration induced by action chunks, which spreads exploration around over a broader region of the state-space. As the critic improves, the actor receives more informative advantage estimates and generates a more stable grasp action sequence. In terms of efficiency, ACPPO also benefits from fewer policy forward passes (one every $h$ steps), which contributes to a 12.7% speedup in the training process. Qualitatively, ACPPO rollouts exhibit more stable grasp poses as displayed in Fig. 2.

## 6.2 PPO WITH REDUCED DECISION FREQUENCY

To isolate the contribution of our action chunking mechanism, we evaluate a naive action repetition variant of PPO. At each boundary time $t$, the policy $\pi_\theta(\cdot \mid s_t)$ samples a single action $a_t$. The action is then duplicated for the next $h - 1$ environment steps,

$$a_{t+k} \leftarrow a_t \quad \text{for } k = 1, \ldots, h - 1,$$

reducing the decision frequency by a factor of $h$. The reward for the chunk boundary is aggregated as:

$$R_t^{(h)} = r_t + \gamma r_{t+1} + \cdots \gamma^{h-1} r_{t+h-1}, \tag{13}$$

and is fed into GAE with the discount factors exponentiated by a factor of $h$.

However, this variant diverges early in training, as action duplication amplifies exploration in unstructured ways, driving the policy into irrelevant regions rather than those useful for dexterous grasping.

## 6.3 EFFECT OF ACTION CHUNK SIZE

We ablate the effect of action chunk sizes, $h = 1, 2, 3, 4, 8$. Since the backward pass of the learning algorithm loop occurs every 8 iterations, it is impractical to use a chunk size that is greater than 8. Furthermore, when using $h = 3$, the last action of the third chunk is discarded.

Table 4 summarizes the results with the best performance at $h = 2$. Increasing the chunk size yields a success rate drop of near 2 3 percentage points across splits, demonstrating the trade-off between temporal coherence and decision frequency. For $h = 8$, learning collapses: a single decision per loop is too coarse, making it impossible to collect meaningful exploration.

Capacity also matters. In our implementation, the output dimension of the chunked actor scales linearly with $h$, while the backbone remains a 4-layer MLP with widths $[1024, 1024, 512, 512]$ for a fair comparison. Larger $h$ increases the number of parameters of the policy, and optimization becomes much more challenging without an increase in the layer dimensions.

Table 4: Ablation study on the chunk size $h$.

| Method | Train (%) | Test (%) | | Training Time (s) |
|--------|-----------|----------|--------|-------------------|
| | | Seen | Unseen | |
| PPO (h=1) | 90.5 | 90.1 | 92.3 | 17803 |
| ACPPO (h=2) | **93.4** | **93.1** | **94.3** | 15,441 |
| ACPPO (h=3) | 91.7 | 91.4 | 91.0 | 15,182 |
| ACPPO (h=4) | 91.4 | 91.1 | 91.1 | 14,997 |
| ACPPO (h=8) | 0 | 0 | 0 | **14,818** |

## 6.4 APPLICATION TO OTHER ENVIRONMENTS

One important question is whether ACPPO is a direct improvement over PPO, analogous to the relationship between TRPO and PPO. Unfortunately, this is not the case. ACPPO introduces a fundamental trade-off between decision frequency and temporal coherence, making its effectiveness highly dependent on the characteristics of the environment.

In universal dexterous grasping, the gain from temporal coherence is advantageous, while high-frequency control is less critical. Before contact with the object, small deviations rarely cause immediate failure, making the task robust to minor errors. Once a stable grasp is achieved, the primary objective shifts to simply maintaining the current pose, which further reduces the need for rapid corrective actions. Overall, the temporal coherence of ACPPO helps improve exploration, without sacrificing performance despite the reduced action frequency.

However, for more dynamic and unstable tasks such as robot balancing or locomotion, common in benchmarks such as Raffin (2020), high-frequency feedback control is crucial. In such systems, even small perturbations can lead to critical failures before ACPPO reaches the next decision step and make recovery impossible when the actor reaches its next decision step. Therefore, ACPPO should not be viewed as a universal replacement for PPO, but rather as a specialized algorithm for domains like universal dexterous grasping, where the benefits of temporally coherent actions outweigh the cost of a lower decision frequency.

## 7 CONCLUSION

We propose an on-policy algorithm that predicts short action sequences using a simple state-value critic and chunked importance sampling, named Action Chunking Proximal Policy Optimization (ACPPO). By moving to the chunked actor and reformulating the PPO surrogate over action chunks, ACPPO delivers temporally coherent exploration, exposing the critic to a broader, more informative subset of the state space, leading to an improved policy. On DexGraspNet, ACPPO outperforms all prior PPO-based methods across train, seen/unseen splits, and training time.

While our study focuses on state-based policies, distillation into vision-based policies via DAgger (Ross et al., 2011) is feasible and could extend our research to sim-to-real transfer. Furthermore, one key limitation of our work is that action chunking can reduce reactivity to sudden perturbations inside a chunk. While perturbations are not a major concern in simulated universal dexterous grasping environments, this can emerge as a major problem when adapting to real, dynamic settings. Future work could extend our approach to adaptive/learned chunk lengths, integrating curriculum or residual controllers, and real-to-sim transfer with tactile sensing to address this issue.

### USE OF LLMS

In this work, Large Language Models GPT 5 Thinking and Gemini 2.5 Pro assisted with polishing writing and conducting literature review.

### ETHICS STATEMENT

We affirm compliance with the ICLR Code of Ethics (https://iclr.cc/public/CodeOfEthics). This work presents Action Chunking Proximal Policy Optimization, an on-

line action chunking reinforcement learning algorithm designed for universal dexterous grasping. All experiments utilized publicly available datasets, without private data.

## REPRODUCIBILITY STATEMENT

We have prioritized reproducibility by detailing the algorithm of ACPPO, including its mathematical foundations, implementation details, and hyperparameters in the main text and appendix. All datasets are publicly accessible and properly cited. Evaluation protocols and metrics are fully described, and the source code is provided in the supplementary materials.

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

# A   MATHEMATICAL DERIVATIONS

## A.1   DERIVATION OF PROXIMAL POLICY OPTIMIZATION

Define the importance ratio $\rho_t(\theta)$ and advantage as $A^\pi(s_t, a_t)$:

$$\rho_t(\theta) := \frac{\pi_\theta(a_t|s_t)}{\pi_{\theta_0}(a_t|s_t)}, \tag{14}$$

$$A^\pi(s_t, a_t) := Q^\pi(s_t, a_t) - V^\pi(s_t). \tag{15}$$

The relative objective function $\mathcal{J}(\theta) - \mathcal{J}(\theta_0)$ for the policy gradient can be formulated as follows:

$$\mathcal{J}(\theta) - \mathcal{J}(\theta_0) = \mathbb{E}_{\tau \sim (p_0, \pi_\theta, p)} \left[ \sum_{t=0}^{T-1} \gamma^t r_t + \gamma V^{\pi_{\theta_0}}(s_{t+1}) - V^{\pi_{\theta_0}}(s_t) \right] \tag{16}$$

$$= \mathbb{E}_{\tau \sim (p_0, \pi_\theta, p)} \left[ \sum_{t=0}^{T-1} \gamma^t (Q^{\pi_{\theta_0}}(s_t, a_t) - V^{\pi_{\theta_0}}(s_t)) \right] \tag{17}$$

$$= \mathbb{E}_{\substack{\tau \sim (p_0, \pi_\theta, p) \\ a_t' \sim \pi_{\theta_0}}} \left[ \sum_{t=0}^{T-1} \gamma^t \frac{\pi_\theta(a_t'|s_t)}{\pi_{\theta_0}(a_t'|s_t)} A^{\pi_{\theta_0}}(s_t, a_t') \right]. \tag{18}$$

Since the expectation that depends on the current policy $\theta$ is hard to utilize, define the surrogate objective function:

$$\mathcal{K}(\theta; \theta_0) = \mathbb{E}_{\tau \sim (p_0, \pi_{\theta_0}, p)} \left[ \sum_{t=0}^{T-1} \gamma^t \rho_t(\theta) A^{\pi_{\theta_0}}(s_t, a_t') \right]. \tag{19}$$

This surrogate objective function is accurate up to the first order, i.e.:

$$\nabla_\theta \mathcal{J}(\theta) \Big|_{\theta=\theta_0} = \nabla_\theta \mathcal{K}(\theta; \theta_0) \Big|_{\theta=\theta_0}. \tag{20}$$

PPO maximizes the surrogate objective function while keeping the current policy $\theta$ close to the driving policy $\theta_0$ by clipping the importance ratio. The corresponding objective function can be written as:

$$\mathcal{L}_{\text{clip}}(\theta) := \mathbb{E}_{\pi_{\theta_0}} \left[ \mathcal{C}_\epsilon \left( \rho_t(\theta), A_t^{\pi_{\theta_0}} \right) \right]. \tag{21}$$

While using $A^{\pi_{\theta_0}}$ directly in the PPO term has no bias, it causes variance as a tradeoff. Therefore, we use a Generalized Advantage Estimate (GAE) $\hat{A}_t^{\text{GAE}(\lambda)}$ with temporal-difference residuals $\delta_t$:

$$\delta_t^{(\phi)} := r_t + \gamma V_\phi(s_{t+1}) - V_\phi(s_t), \tag{22}$$

$$\hat{A}_t^{\text{GAE}(\lambda)} := \sum_{l=0}^{\infty} (\gamma\lambda)^l \delta_{t+l}. \tag{23}$$

The GAE estimators are now biased, but have smaller variance compared to $A^{\pi_{\theta_0}}$. Using GAE, we finalize the PPO objective function as:

$$\mathcal{L}_{\text{PPO}}(\theta) := \mathbb{E}_{\pi_{\theta_0}} \left[ \mathcal{C}_\epsilon \left( \rho_t(\theta), \hat{A}_t^{\text{GAE}(\lambda)} \right) \right]. \tag{24}$$

For the implementation of a continuous action space, the policy is commonly modeled as a Gaussian distribution. This allows the agent to sample stochastic actions from a continuous action space, and also keeps $\pi_\theta(a_t|s_t)$ for calculating the importance sampling ratio.

$$\pi_\theta(a_t|s_t) = \frac{1}{\sqrt{(2\pi)^{|A|}|\Sigma|}} \exp\left( -\frac{1}{2}(a_t - \mu_\theta(s_t))^T \Sigma_\theta(s_t)^{-1} (a_t - \mu_\theta(s_t)) \right). \tag{25}$$

## A.2 ACTION CHUNKING PPO

Define the chunked importance sampling ratio as:

$$\rho_{t,h}^{ch}(\theta) := \frac{\pi_\theta(a_{t:t+h-1}|s_t)}{\pi_{\theta_0}(a_{t:t+h-1}|s_t)}. \tag{26}$$

Starting from 17, we apply a $h$-step tower property:

$$\mathbb{E}_{\tau \sim (p_0, \pi_\theta, p)}\left[\sum_{t=0}^{T-1} \gamma^t (Q^{\pi_{\theta_0}}(s_t, a_t) - V^{\pi_{\theta_0}}(s_t))\right]$$

$$= \mathbb{E}_{\tau \sim (p_0, \pi_\theta, p)}\left[\sum_{t=0}^{T-1} \gamma^t A^{\pi_{\theta_0}}(s_t, a_t)\right]$$

$$= \mathbb{E}_{\tau^{(h)}}\left[\sum_{t=0}^{h-1} \gamma^t A^{\pi_{\theta_0}}(s_t, a_t) + \mathbb{E}_{\pi_\theta}\left[\sum_{t=h}^{T-1} \gamma^t A^{\pi_{\theta_0}}(s_t, a_t)\Big|\tau^{(h)}\right]\right]$$

$$= \mathbb{E}_{\substack{s_0 \sim p_0, s_{t+1} \sim p(\cdot|s_t, a_t) \\ a_{0:h-1} \sim \pi_\theta(\cdot|s_0) \\ a'_{0:h-1} \sim \pi_{\theta_0}(\cdot|s_0)}}\left[\frac{\pi_\theta(a'_{0:h-1}|s_0)}{\pi_{\theta_0}(a'_{0:h-1}|s_0)}\sum_{t=0}^{h-1} \gamma^t A^{\pi_{\theta_0}}(s_t, a'_t) + \mathbb{E}_{\pi_\theta}\left[\sum_{t=h}^{T-1} \gamma^t A^{\pi_{\theta_0}}(s_t, a_t)\Big|\tau^{(h)}\right]\right]$$

$$= \mathbb{E}_{\substack{\tau \sim (p_0, \pi_\theta, p) \\ a'_t \sim \pi_{\theta_0}}}\left[\sum_{l=0}^{(T-1)/h}\left[\rho_{lh,h}^{ch}(\theta)\sum_{t=lh}^{(l+1)h-1} \gamma^t A^{\pi_{\theta_0}}(s_t, a'_t)\right]\right]$$

This formula is exact, regarding that all actions inside the chunk are being conditioned on the same state, independent of each other. However, simply using the cumulative discounted advantage $\sum \gamma^t A(s,a)$ causes a high variance. Instead, we define a chunked advantage:

$$A^{\pi_{\theta_0}}(s_t, a_{t:t+h-1}) = \sum_{k=0}^{h-1} \gamma^k r_{t+k} + \gamma^h V^{\pi_{\theta_0}}(s_{t+h}) - V^{\pi_{\theta_0}}(s_t). \tag{27}$$

Note that

$$\mathbb{E}_p\left[\sum_{k=0}^{h-1} \gamma^k A^{\pi_{\theta_0}}(s_{t+k}, a_{t+k})\right] = \mathbb{E}_p\left[A_h^{\pi_{\theta_0}}(s_t, a_{t:t+h-1})\right].$$

Using the chunked advantage, we can rewrite:

$$\mathbb{E}_{\substack{\tau \sim (p_0, \pi_\theta, p) \\ a'_t \sim \pi_{\theta_0}}}\left[\sum_{l=0}^{(T-1)/h}\left[\rho_{lh,h}^{ch}(\theta)\sum_{t=lh}^{(l+1)h-1} \gamma^t A^{\pi_{\theta_0}}(s_t, a'_t)\right]\right]$$

$$= \mathbb{E}_{\substack{\tau \sim (p_0, \pi_\theta, p) \\ a'_t \sim \pi_{\theta_0}}}\left[\sum_{l=0}^{(T-1)/h} \gamma^{lh} \rho_{lh,h}^{ch}(\theta) A_h^{\pi_{\theta_0}}(s_{lh}, a_{lh:lh+h-1})\right]$$

$$\approx \mathbb{E}_{\tau \sim (p_0, \pi_{\theta_0}, p)}\left[\sum_{l=0}^{(T-1)/h} \gamma^{lh} \rho_{lh,h}^{ch}(\theta) A_h^{\pi_{\theta_0}}(s_{lh}, a_{lh:lh+h-1})\right]$$

The final $\approx$ is exactly the approximation we make for the surrogate objective function in 19. For further reducing the variance, we replace the chunked advantage with the same GAE we use for PPO:

$$\mathcal{L}_{\text{ACPPO}}^h(\theta) := \mathbb{E}_t\left[\mathcal{C}_\epsilon\left(\rho_{t,h}^{chunk}(\theta), \hat{A}_t^{\text{GAE}(\lambda)}\right)\right]. \tag{28}$$

While the application of GAE introduces additional bias into the surrogate loss term, this bias is smaller compared to the case of PPO. With a chunk size of $h$, the GAE can be decomposed as:

$$\hat{A}_t^{\text{GAE}(\lambda)} = \underbrace{\sum_{j=0}^{h-1} (\gamma\lambda)^j \delta_{t+j}}_{\text{inside chunk}} + \underbrace{\sum_{j=h}^{\infty} (\gamma\lambda)^j \delta_{t+j}}_{\text{tail bias}}. \tag{29}$$

In PPO, the tail bias term starts at $j = 1$, while for ACPPO, the tail bias term starts at $j = h(> 1)$. This suppresses the bias for ACPPO by a factor of $(\gamma\lambda)^{h-1}$ relative to PPO, yielding a more accurate advantage estimate.

### A.3 DERIVATION OF ACTION CHUNKED IMPORTANCE SAMPLING

We further elaborate the details on applying importance sampling to the relative objective function in the case of $h = 2$. Starting from the relative objective function, all formulations are exact.

$$\mathcal{J}(\theta) - \mathcal{J}(\theta_0)$$

$$= \mathbb{E}_{\tau \sim (p_0, \pi_\theta, p)} \left[ \sum_{t=0}^{T-1} \gamma^t r_t + \gamma V^{\pi_{\theta_0}}(s_{t+1}) - V^{\pi_{\theta_0}}(s_t) \right]$$

$$= \mathbb{E}_{\tau \sim (p_0, \pi_\theta, p)} \left[ \sum_{t=0}^{T-1} \gamma^t (Q^{\pi_{\theta_0}}(s_t, a_t) - V^{\pi_{\theta_0}}(s_t)) \right]$$

$$= \mathbb{E}_{\tau \sim (p_0, \pi_\theta, p)} \left[ \sum_{t=0}^{T-1} \gamma^t A^{\pi_{\theta_0}}(s_t, a_t) \right]$$

$$= \mathbb{E}_{\tau^{(h)}} \left[ \sum_{t=0}^{h-1} \gamma^t A^{\pi_{\theta_0}}(s_t, a_t) + \mathbb{E}_{\pi_\theta} \left[ \sum_{t=h}^{T-1} \gamma^t A^{\pi_{\theta_0}}(s_t, a_t) \middle| \tau^{(h)} \right] \right]$$

$$= \mathbb{E}_{\substack{(a_0,a_1) \sim \pi_\theta(\cdot|s_0) \\ s_1 \sim p(\cdot|s_0,a_0)}} \left[ A(s_0, a_0) + \gamma A(s_1, a_1) + \mathbb{E}[\cdots] \right]$$

$$= \sum_{(a_0,a_1) \in A^2, s_1 \in S} \left[ \pi_\theta(a_0, a_1|s_0) \Big( A(s_0, a_0) + \gamma p(s_1|s_0, a_0) A(s_1, a_1) \Big) + \mathbb{E}[\cdots] \right]$$

$$= \sum_{(a_0,a_1) \in A^2, s_1 \in S} \left[ \pi_{\theta_0}(a_0, a_1|s_0) \frac{\pi_\theta(a_0, a_1|s_0)}{\pi_{\theta_0}(a_0, a_1|s_0)} \Big( A(s_0, a_0) + \gamma p(s_1|s_0, a_0) A(s_1, a_1) \Big) + \mathbb{E}[\cdots] \right]$$

$$= \mathbb{E}_{\substack{(a_0,a_1) \sim \pi_{\theta_0}(\cdot|s_0) \\ s_1 \sim p(\cdot|s_0,a_0)}} \left[ \frac{\pi_\theta(a_0, a_1|s_0)}{\pi_{\theta_0}(a_0, a_1|s_0)} \Big( A(s_0, a_0) + \gamma A(s_1, a_1) \Big) + \mathbb{E}[\cdots] \right]$$

$$= \mathbb{E}_{\substack{\tau \sim (p_0, \pi_\theta, p) \\ a_t \sim \pi_{\theta_0}}} \left[ \sum_{l=0}^{(T-1)/2} \left[ \frac{\pi_\theta(a_l, a_{l+1}|s_l)}{\pi_{\theta_0}(a_l, a_{l+1}|s_l)} \Big( A(s_l, a_l) + \gamma A(s_{l+1}, a_{l+1}) \Big) \right] \right]$$

## B IMPLEMENTATION DETAILS

### B.1 REWARD FUNCTION

The reward function for training the policy is defined as:

$$R = R_{\text{dist}} + R_{\text{align}} + f_{\text{contact}} \left( R_{\text{goal}} + R_{\text{lift}} + R_{\text{bonus}} \right).$$

$R_{dist}$ penalizes the object-hand distance, encouraging the hand to reach for the object before contact. Defining $X_{obj}$, $X_{hand}$, $X_{finger,i}$ as the position of the object, the palm, and the $i$th finger respectively, $R_{dist}$ is defined as follows:

$$R_{dist} = -1.0 \times ||X_{obj} - X_{hand}||_2 - 0.5 \times \sum_{i=1}^{5} ||X_{obj} - X_{finger,i}||_2.$$

$R_{align}$ encourages the thumb to spread away from the other four fingers for the stability of grasping. Let the unit vectors from the object to the palm and each fingertips be:

$$\mathbf{u}_{\text{palm}} = \frac{X_{\text{hand}} - X_{\text{obj}}}{||X_{\text{hand}} - X_{\text{obj}}||_2}, \qquad \mathbf{u}_{\text{finger},i} = \frac{X_{\text{finger},i} - X_{\text{obj}}}{||X_{\text{finger},i} - X_{\text{obj}}||_2}.$$

Defining pairwise cosines $c_{i,j} = \mathbf{u}_{\text{finger},i}^{\top}\mathbf{u}_{\text{finger},j}$, the alignment reward $R_{\text{align}}$ is:

$$R_{\text{align}} = -\lambda_{\text{align}}\Big(c_{2,1} + c_{3,1} + 0.5c_{4,1} + 0.1c_{5,1}\Big).$$

The remaining rewards activate only once the hand is close enough to the object:

$$f_{\text{contact}} = \mathbb{1}[\,\|X_{\text{obj}} - X_{\text{hand}}\|_2 \leq 0.12\,] \cdot \mathbb{1}\left[\sum_{i=1}^{5} \|X_{\text{obj}} - X_{\text{finger},i}\|_2 \leq 0.06\right].$$

where $\mathbb{1}$ is the indicator function.

$R_{\text{goal}}$ is the term that drives the object closer to the goal:

$$R_{\text{goal}} = 0.9 - 2\,\|X_{\text{obj}} - X_{\text{goal}}\|_2.$$

$R_{\text{lift}}$ encourages lifting the object after contact, with staged bonuses at increasing height. Defining $Z_{\text{min}}(X_{\text{obj}})$ as the object's lowest point and $a_z$ as the action's vertical component, the lifting reward $R_{\text{lift}}$ is formulated as:

$$R_{\text{lift}} = \mathbb{1}[\,Z_{\text{min}}(X_{\text{obj}}) \geq 0.63\,]\,(0.1 + 0.1\,a_z) + \mathbb{1}[\,Z_{\text{min}}(X_{\text{obj}}) \geq 0.80\,]\,0.2.$$

Finally, $R_{\text{bonus}}$ is a smooth shaping near the goal for precise placement:

$$R_{\text{bonus}} = \mathbb{1}[\,\|X_{\text{obj}} - X_{\text{goal}}\|_2 \leq 0.05\,]\;\frac{1}{1 + 10\,\|X_{\text{obj}} - X_{\text{goal}}\|_2}.$$

### B.2 HYPERPARAMETERS

For reproducibility, we state our hyperparameters for ACPPO as follows. The hyperparameters for PPO are identical, with only setting $h = 1$.

Table 5: Hyperparameters of ACPPO.

| Name | Symbol | Value |
|---|---|---|
| Episode length | -- | 200 |
| Rollout steps per iteration | -- | 8 |
| Training epochs per iteration | -- | 5 |
| Num. minibatches per epoch | -- | 4 |
| MLP dimensions | -- | [1024, 1024, 512, 512] |
| Optimizer | -- | Adam (Kingma & Ba, 2015) |
| Nonlinearity | -- | ELU (Clevert et al., 2015) |
| Clip gradient norm | -- | 1.0 |
| Desired KL | -- | 0.016 |
| Initial noise std. | -- | 0.8 |
| Clip observations | -- | 5.0 |
| Clip actions | -- | 1.0 |
| Chunk size | $h$ | 2 |
| Learning rate | $\eta$ | 3e-4 |
| Discount factor | $\gamma$ | 0.96 |
| GAE lambda | $\lambda$ | 0.95 |
| Clip range | $\epsilon$ | 0.2 |

The hyperparameters for ACFQL are as follows.

Table 6: Hyperparameters of ACFQL.

| Name | Symbol | Value |
|---|---|---|
| Episode length | -- | 200 |
| Rollout steps per iteration | -- | 8 |
| Training epochs per iteration | -- | 5 |
| Num. minibatches per epoch | -- | 4 |
| MLP dimensions | -- | [1024, 1024, 512, 512] |
| Optimizer | -- | Adam (Kingma & Ba, 2015) |
| Nonlinearity | -- | ELU (Clevert et al., 2015) |
| Clip gradient norm | -- | 1.0 |
| Initial noise std. | -- | 0.8 |
| Clip observations | -- | 5.0 |
| Clip actions | -- | 1.0 |
| Chunk size | $h$ | 5 |
| Learning rate | $\eta$ | 2e-4 |
| Discount factor | $\gamma$ | 0.99 |
| Clip range | $\epsilon$ | 0.2 |

