# OpenReview forum: "Action Chunking Proximal Policy Optimization for Universal Robotic Dexterous Grasping"
_ICLR.cc/2026/Conference — ICLR 2026 Conference Withdrawn Submission_

### Official Review · Reviewer_nFS9 · 2025-10-20

**Soundness:** 2
**Presentation:** 2
**Contribution:** 1
**Rating:** 2
**Confidence:** 4

**Summary:**

This paper proposes a reinforcement learning algorithm, Action Chunking Proximal Policy Optimization (ACPPO), for general-purpose grasping with high-degree-of-freedom robotic hands. The method extends PPO by incorporating action chunking, which outputs multiple actions at each step, enabling temporally coherent exploration without introducing high-dimensional Q-functions. Experiments on the DexGraspNet dataset demonstrate that ACPPO outperforms existing PPO-based methods, achieving higher success rates (95.4%) and faster training (2.3× speedup).

**Strengths:**

1. The paper provides a comprehensive literature review and includes a reasonably broad selection of baseline methods for comparison.
2. The ablation studies are comprehensive, including experiments on chunk size, decision frequency, and comparisons with ACFQL, which collectively validate the design choices.

**Weaknesses:**

1. The contribution of this work is rather incremental. The method mainly incorporates action chunking into PPO, without offering substantial conceptual novelty or broader insights.
2. The paper lacks sufficient depth and clarity. The presentation is somewhat disorganized, making it difficult to follow the main ideas and contributions.
3. The improvement in performance is modest: ACPPO achieves only a slight increase over the best baseline (ResDex, 94.6% → 95.4%). The main contribution lies in simplifying the training process rather than delivering a substantial performance gain.

**Questions:**

1. The ablation study indicates that the optimal chunk size is $ h=2 $, which seems to contradict the claimed benefits of action chunking. Given that the typical control frequency in the previous work is $ 60 \mathrm{Hz} $, a chunk size of two steps corresponds to a very short time interval.
2. The reported success rate of vanilla PPO (over 90%) is inconsistent with previous works[1].
3. In the supplementary videos, the simulator still exhibits noticeable jitter, and there is little improvement in motion coherence, which does not align with the authors’ claims.
4. Although the paper claims that action chunking encourages temporally coherent exploration, the comparison with vanilla PPO shows only marginal improvement in training efficiency.


[1] Yinzhen Xu, et al. Unidexgrasp: Universal robotic dexterous grasping via learning diverse proposal generation and goal-conditioned policy, CVPR 2023

---

### Official Review · Reviewer_Ybn2 · 2025-10-26

**Soundness:** 3
**Presentation:** 3
**Contribution:** 3
**Rating:** 4
**Confidence:** 3

**Summary:**

This paper proposes an action chunking framework for learning universal dexterous grasping, and introduces the use of a state value function instead of an action value function to mitigate the challenge posed by extremely high-dimensional action spaces. Comprehensive simulation experiments are conducted to demonstrate the effectiveness of the proposed design.

**Strengths:**

1. The proposed method achieves a success rate comparable to the state of the art, without relying on residual policies or curriculum learning, and demonstrates significantly higher efficiency than other approaches.
2. A comprehensive ablation study is conducted to validate the effectiveness of the proposed method.

**Weaknesses:**

1. In other papers, PPO typically performs poorly; however, in this work, PPO achieves even higher performance than PPO with curriculum reported elsewhere. If the PPO implementation in this paper is inherently better than in previous works, the additional improvement provided by the proposed method may not appear significant. Could the authors clarify the reason behind this?
2. In the supplementary demonstrations, the dexterous hand appears to oscillate after grasping objects, suggesting that using action chunking may not clearly reduce jittering.

**Questions:**

See weakness.

---

### Official Review · Reviewer_FmBk · 2025-10-27

**Soundness:** 1
**Presentation:** 3
**Contribution:** 2
**Rating:** 2
**Confidence:** 4

**Summary:**

The paper studies PPO with an action-chunking actor for learning universal dexterous grasping policies. Experiments on DexGraspNet show that the method outperforms both previous RL-based methods and the vanilla PPO without action chunking.

**Strengths:**

- RL with action chunking is an important problem in robot learning, as imitation learning with action chunking has shown great promise.
- The choice of PPO for action-chunking RL makes sense, avoiding the need to learn the high-dimensional Q-function.
- The algorithm achieves very good performance on DexGraspNet, surpassing prior RL methods.

**Weaknesses:**

- The good result of vanilla PPO without any curriculum learning design (achieving 90+% success rates as shown in Table 3 and 4) is strange. Prior works (UniDexGrasp, UniDexGrasp++, and ResDex) show that directly running PPO on 3200 DexGraspNet objects results in poor success rates (typically 10%~60%) due to the inherent multi-task optimization challenge.
- Even if the results are reliable, the improvement of PPO with action chunking compared with vanilla PPO is quite minor (from 90% to 93%). Simply increasing the training time of vanilla PPO may leads to the same gain.
- The claim that "ACPPO improves exploration with temporally coherent, smooth actions" (line 311-318) may not be true. Since the policy parameters are trained from scratch and the Gaussian randomness for each action dimension is independent, the policy is not encouraged to output temporally coherent actions no matter action chunking is used. The cited works (line 307-310) are either imitation learning or offline-to-online RL works and their policies are initially trained from high-quality demonstrations where actions are smooth and temporally coherent. Therefore, these works cannot induce the same claim in this paper under the RL-from-scratch setting.
- Experimental results are not comprehensive. Though action-chunking may not succeed in locomotion tasks, the ACPPO proposed in this paper is not specifically designed for the grasping task. As a general purpose algorithm design, evaluation on more tasks (such as manipulation tasks in the Meta-World benchmark) is quite necessary. On the other side, if the paper focuses on the specific dexterous grasping task, it is necessary to include vision-based results, compare with more recent methods like RobustDexGrasp, and do some sim-to-real experiments.

**Questions:**

- Please see my concerns in Weaknesses.
- The issue of being less reactive of action-chunking policies can be addressed by receding horizon control during inference (e.g. predict the whole action chunk but only execute the first one). Can this be integrated into the RL algorithm to address dynamic tasks such as locomotion?

---

### Official Review · Reviewer_j8J7 · 2025-10-28

**Soundness:** 3
**Presentation:** 2
**Contribution:** 2
**Rating:** 4
**Confidence:** 4

**Summary:**

This paper proposes **Action Chunking PPO (ACPPO)** for high-DoF dexterous grasping, where standard PPO fails due to poor exploration. The authors identify that prior action chunking (ACRL) methods are intractable in this domain because they require learning a high-dimensional $Q(s, a_{chunk})$. ACPPO avoids this by modifying the PPO objective to use a **chunked importance sampling (IS) ratio** $\rho_{t,h}^{ch}(\theta)$ while retaining a simple $V(s)$ critic and standard GAE. This novel on-policy formulation achieves state-of-the-art results on DexGraspNet, training 2.3x faster than prior work without auxiliary augmentations.

**Strengths:**

***Novel Problem-Solving:** The paper clearly identifies the $Q(s, a_{chunk})$ bottleneck that makes prior ACRL methods infeasible for high-DoF robotics and proposes an elegant on-policy solution.

***State-of-the-Art Performance:** ACPPO achieves SOTA results on the complex DexGraspNet benchmark.

***Exceptional Training Efficiency:** The 2.3x training speedup, achieved without any auxiliary mechanisms, highlights the efficiency of the core algorithmic change.

***Strong Ablation Study:** The ablation showing that simple action repetition diverges effectively proves the contribution is the chunked optimization, not just a lower decision frequency.

**Weaknesses:**

* **Mismatch Between Motivation and Results:** The central premise of the paper is that action chunking provides temporally coherent exploration. However, the empirical results (Table 4) show that the best performance is achieved with a minimal chunk size of $h=2$. Performance degrades at $h=3$ and $h=4$, and collapses to 0% at $h=8$. A chunk of 2 barely qualifies as "temporally coherent" and undermines the core motivation. The paper would be far more convincing if it could demonstrate SOTA performance with a more substantial chunk length (e.g., $h \ge 5$).

* **Limited Methodological Novelty:** The proposed change, while elegant, is a single modification to the PPO objective. The core components (actor-critic architecture, $V(s)$ critic, GAE) are identical to PPO. The contribution is essentially replacing the per-step IS ratio $\rho_t$ with a chunked IS ratio $\rho_{t,h}^{ch}$. For an ICLR submission, this level of technical contribution is borderline.

* **Shallow Analysis of Failure Mode:** The paper's analysis of the $h=8$ failure is shallow, attributing it to a "coarse" decision frequency (a behavioral explanation). A more critical optimization-based analysis is missing. A far more likely culprit is the **instability of the chunked importance sampling ratio**. The variance of an IS ratio grows exponentially with the sequence length ($h$). It is highly probable that for $h=8$, the ratio $\rho_{t,h=8}^{ch}$ is so volatile that it is *always* outside the PPO clipping range $[1-\epsilon, 1+\epsilon]$. This would cause the clipping mechanism to **zero out the gradient signal**, making learning impossible. This suggests that on-policy, IS-based methods like PPO may be fundamentally unsuited for large chunk sizes—a critical limitation that is not addressed.

* **Unverified Theoretical Claims:** The authors claim (Eq. 12) that their formulation reduces the "tail bias" from GAE. This is a key justification for their specific objective function, yet it is presented without any empirical validation. An experiment measuring the advantage estimation bias (e.g., relative to Monte Carlo returns) for PPO vs. ACPPO would be required to substantiate this claim.

* **Missing Diagnostic Experiments:** Given the hypothesis about the clipping mechanism, a crucial experiment is missing: a plot of the
 **percentage of clipped updates** as a function of the chunk size $h$. This single experiment would provide vital insight into the algorithm's optimization dynamics and likely confirm why $h=8$ fails.

* **Limited Scope:** The empirical validation is confined to a single, albeit difficult, task family. The authors admit in Section 6.4 that the method would likely fail in dynamic tasks (e.g., locomotion), limiting its generality.

**Questions:**

1.  How do the authors reconcile the "temporal coherence" motivation with the empirical fact that the minimal chunk $h=2$ is optimal?

2.  Please provide a plot of the **percentage of clipped updates** vs. chunk size $h$ (from $h=1$ to $h=8$). Does this confirm the hypothesis that the $h=8$ gradient is zeroed out by the clipping mechanism?

3.  Can the authors provide empirical evidence for their claim (Eq. 12) that ACPPO reduces GAE's "tail bias"? A direct measurement of advantage estimation error would be convincing.

4.  Given the IS variance issue, do the authors believe that on-policy, IS-based methods like PPO are fundamentally a poor choice for action chunking with $h > 2$, and that this problem requires an off-policy formulation?

---

### Note · Authors · 2025-11-13

I have read and agree with the venue's withdrawal policy on behalf of myself and my co-authors.